# Differences in Illness Severity among Circulating Norovirus Genotypes in a Large Pediatric Cohort with Acute Gastroenteritis

**DOI:** 10.3390/microorganisms8121873

**Published:** 2020-11-26

**Authors:** Sudha Bhavanam, Stephen B. Freedman, Bonita E. Lee, Ran Zhuo, Yuanyuan Qiu, Linda Chui, Jianling Xie, Samina Ali, Otto G. Vanderkooi, Xiaoli L. Pang

**Affiliations:** 1Department of Laboratory Medicine and Pathology, University of Alberta, Edmonton, AB T6G 2R3, Canada; bhavanam@ualberta.ca (S.B.); rzhuo@ualberta.ca (R.Z.); yuanyuan@ualberta.ca (Y.Q.); Linda.Chui@albertaprecisionlabs.ca (L.C.); 2Divisions of Pediatric Emergency Medicine and Gastroenterology, Departments of Pediatrics and Emergency Medicine, Alberta Children’s Hospital, Alberta Children’s Hospital Research Institute, Cumming School of Medicine, University of Calgary, Calgary, AB T3B 6A8, Canada; stephen.freedman@albertahealthservices.ca; 3Department of Pediatrics, Faculty of Medicine & Dentistry, Women and Children’s Health Research Institute, Stollery Children’s Hospital, University of Alberta, Edmonton, AB T6G 2R3, Canada; bonita.lee@albertahealthservices.ca (B.E.L.); sali@ualberta.ca (S.A.); 4Public Health Laboratories (ProvLab), Alberta Precision Laboratories (APL), Edmonton, AB T6G 2J2, Canada; 5Departments of Pediatrics, Cumming School of Medicine, University of Calgary, Calgary, AB T3B 6A8, Canada; jianling.xie@albertahealthservices.ca; 6Departments of Pediatrics, Microbiology, Immunology and Infectious Diseases, Pathology & Laboratory Medicine and Community Health Sciences, Alberta Children’s Hospital, Alberta Children’s Hospital Research Institute, Cumming School of Medicine, University of Calgary, Calgary, AB T3B 6A8, Canada; ovanderk@ucalgary.ca; 7University of Alberta, Edmonton, AB T6G 2R3, Canada; 8University of Calgary, Calgary, AB T3B 6A8, Canada; 9Alberta Health Services, Edmonton, AB T5J 3E4, Canada

**Keywords:** norovirus, gastroenteritis, clinical severity, genotypes, children

## Abstract

Norovirus is a major pathogen identified in children with acute gastroenteritis (AGE), little is known about the strain’s diversity and their clinical severity. Stool and/or rectal swabs were collected from children ≤18 years of age recruited at emergency departments (ED), and a provincial nursing advice phone line due to AGE symptoms in the province of Alberta, Canada between December 2014 and August 2018. Specimens were tested using a reverse transcription real time PCR and genotyped by Sanger sequencing. The Modified Vesikari Scale score (MVS) was used to evaluate the disease severity. The objectives are to identify the Genogroup and Genotype distribution and to compare illness severity between the GI and GII genogroups and to complete further analyses comparing the GII genotypes identified. GII.4 was the genotype most commonly identified. Children with GII.4 had higher MVS scores (12.0 (10.0, 14.0; *p* = 0.002)) and more prolonged diarrheal (5 days (3.0, 7.8)) and vomiting (3.2 days (1.7, 5.3; *p* < 0.001)) durations compared to other non GII.4 strains. The predominant strain varied by year with GII.4 Sydney[P31] predominant in 2014/15, GII.4 Sydney[P16] in 2015/16 and 2017/18, and GII.3[P12] in 2016/17. Genogroup II norovirus strains predominated in children with AGE with variance between years; clinical severity associated with different strains varied with episodes being most severe among GII.4 infected children.

## 1. Introduction

Acute gastroenteritis (AGE) is associated with an estimated 500,000 deaths among children younger than five years globally [1]. While noroviruses are currently regarded as being the leading contributor to AGE in people of all ages [2,3], incidence rates are highest among young children [4]. Although most infected individuals have mild illnesses of short duration characterized predominantly by vomiting [5], infants more often have severe disease and are more likely to seek medical care and be hospitalized [6].

Noroviruses are a genetically diverse group of viruses in the *Caliciviridae* family that are classified into seven genogroups (GI-GVII), of which norovirus GI, GII, and GIV infect humans [5]. Globally, viruses of the GII.4 genotype are the leading cause of norovirus disease [7], with new variants emerging every two years to four years [8,9] In addition, the GII.4 genotype has been associated with greater symptom severity and health care resource consumption [10]. Although outbreak reports have been analyzed to quantify hospitalization and mortality rates associated with GII.4 compared with non-GII.4 disease [10], no data has previously characterized illness severity for varying genotypes using commonly employed clinical disease severity measures.

A dual nomenclature system has been proposed for norovirus GI and GII based on VP1 and RNA-dependent RNA polymerase (RdRp) protein sequences [11]. According to the recently updated norovirus classification, the number of genogroups and genotypes were based on complete VP1 amino acids and the number of P-groups and P-types based on a partial region (762 nucleotides) of the RdRp at the 3′-end of Open Reading Frame-1 (ORF-1). Dual typing (ORF1-RdRp = P type, ORF2 = genotype) is now used routinely for the classification for norovirus. Nine GI genotypes and 22 GII genotypes have been recognized in human [12,13].

In children with endemic AGE, the etiologic genotypes vary more than is seen among outbreaks, [14,15,16]. Although a biennial genotype pattern with rapid evolution of different predominant GII.4 strains occurs during norovirus outbreaks in our province (Alberta, Canada), corresponding genotype data related to sporadic disease is lacking [17,18].

To fill these knowledge gaps, we sought to characterize the norovirus strains in a cohort of prospectively enrolled children with AGE. The secondary objectives were to compare illness severity between the GI and GII genogroups and to conduct further analyses comparing the GII genotypes identified.

## 2. Materials and Methods

### 2.1. Study Design

This study was approved by the Research Ethics Boards of the University of Alberta and the University of Calgary. Eligible participants provided informed consent (and assent when appropriate) and were recruited by the Alberta Provincial Pediatric EnTeric Infection TEam (APPETITE) between December 2014 and August 2018 in accordance with the approved study protocol [19].

Eligible cases were ≤18 years of age, presented with ≥3 episodes of vomiting and/or diarrhea in the preceding 24 h, and had been ill for <7 days. Children who could not complete follow-up, those enrolled in the previous 14 days, and those presenting with a mental health related primary concern, neutropenia, or in need of emergent medical intervention were ineligible. Potentially eligible AGE cases were identified in the ED of the Alberta Children’s Hospital (Calgary, AB, Canada) and the Stollery Children’s Hospital (Edmonton, AB, Canada). Additional participants who were deemed not to require emergency treatment were recruited from a provincial telephone nursing advice line (Healthlink Alberta).

### 2.2. Specimen Collection

Two Flocked swabs (FLOQSwab, Copan Italia, Brescia, Italy) were inserted sequentially into the rectum of each participant recruited in the ED. One swab was placed into a dry, sterile tube, and the other was inserted into 2-mL modified Cary-Blair transport media, FecalSwab (Copan Italia, Brescia, Italy). Bulk stool specimens were collected in sterile containers (V302-F, Starplex Scientific Inc., Etobicoke, ON, Canada). If a bulk stool specimen was not provided before ED discharge, caregivers were instructed to collect a specimen at home. Specimens collected at home were stored at room temperature for up to 12 h and then retrieved and transported at 4 °C to the testing laboratory by a study-funded courier in cooler boxes containing ice packs.

For children recruited via HealthLink, stool collection kits containing two rectal swabs, a stool container, and instructions were couriered to the home. Caregivers were asked to collect the first stool sample produced and perform the rectal swabs at the same time following enrollment. Once received at the laboratory, all specimens were frozen immediately at −80 °C until analyzed.

### 2.3. Data Collection

Trained research assistants administered a structured survey to eligible consented caregivers (and child assent when appropriate). Fourteen days following enrollment, participants completed a follow-up survey. Surveys were administered either by telephone or electronically. Those opting for electronic follow-up received daily e-mail reminders for up to 3 days. If the electronic survey was not completed after these reminders, participants were contacted by telephone.

### 2.4. Enteropathogen Testing

Total nucleic acid was extracted from 300 μL of rectal swab PBS-suspension or 100–150 mg solid or 100 μL liquid stool using NucliSENS^®^ easyMag^®^. Nucleic acid extracts were reverse transcribed with Invitrogen SuperScript^®^II and tested. Two molecular approaches were employed. The Luminex xTAG gastrointestinal pathogen panel (GPP; Luminex Molecular Diagnostics, Austin, TX, USA) is a qualitative multiplex molecular-based syndromic panel that identifies nine bacterial targets (*Campylobacter spp*., *Clostridioides* [formerly *Clostridium*] difficile toxin A/B, *E. coli* O157, enterotoxigenic *E. coli* [ETEC], Shiga toxin-producing *E. coli* [STEC] stx1 and stx2, *Salmonella* spp., *Shigella spp*., *Vibrio cholerae*, *Yersinia* spp.) in addition to three viral targets (human adenovirus 40/41 (HAdV 40/41), norovirus GI/GII and group A rotavirus) [20]. All samples were further tested for viruses using an in-house Gastroenteritis Virus Panel (GVP) that uses RT-qPCR on the 7500 Sequence Detection System from Applied Biosystems (ABI, Foster City, CA, USA) [21]. Reverse transcription (RT) was performed using 5 µL of nucleic acid extracts and SuperScript^®^II (Thermo Fisher Scientific, Burlington, ON, Canada) according to the manufacturer’s instructions followed by three simultaneous duplex qPCR reactions (the duplex format: generic adenovirus and rotavirus, astrovirus, norovirus GI and GII and sapovirus) on a single run. We classified a case patient as positive for norovirus GI or GII if any specimen yielded positive results on either assay.

### 2.5. P-C Genotyping with Sanger Sequencing

For each participant with a norovirus positive specimen, either stool or rectal swab was selected for genotyping based on the lower Ct value. Briefly, the nucleic acid extract from each sample was subjected to RT with random primers. The resulting cDNA was PCR amplified in the C region (capsid) using primer pairs G2SKF and G2SKR to amplify positions 5058–5401 (344 bp) for GII genotypes and primer pairs G1SKF and G1SKR to amplify positions 5342–5671 (330 bp) for GI genotypes [22]. For RdRp (P) typing GII positive samples were further subjected to a heminested PCR using the primer pair LV4282-99F and G2SKR and primers LV4282-99F and COG2R to generate amplicons of 1108 bp and 818 bp respectively, in the region spanning RdRp and VP1[23]. All positive PCR products were purified with QIAquick^®^ Gel Extraction kit (Qiagen, Germantown, MD, USA) according to the manufacturer’s instructions. The nucleotide sequences of each sample were subjected to Sanger sequencing using the BigDye Terminator v3.1 Cycle Sequencing Kit (Life Technologies, Austin, TX, USA). Sequence data were analyzed using MEGA X 10.0.5 and the genotypes were assigned using norovirus Genotyping tool.

### 2.6. Clinical Characteristics and the Modified Vesikari Scale Score

Gastroenteritis illness severity was quantified using the Modified Vesikari Scale (MVS) score [24,25]. This scoring system is commonly employed to evaluate rotavirus and norovirus disease severity in children [26]. The score captures duration of diarrhea and vomiting, maximal number of vomiting and diarrheal episodes per 24-h period, maximal body temperature, treatments provided, and healthcare seeking behaviors. In the cases captured through the nursing advice phone line, temperature was provided directly by the caregivers based on the maximal recorded temperature at home. MVS scores were calculated using data that described the course of disease from symptom onset (i.e., before the ED visit) through to symptom resolution (i.e., as reported on 14-day follow-up). Scores were classified as mild (0–8), moderate (9–10), and severe (≥ 11) [25]. Based on the caregiver report, dehydration was quantified using the Clinical Dehydration Scale (CDS) score, with scores of 0, 1–4, and 5–8 indicating none, mild to moderate, and severe dehydration, respectively [27].

### 2.7. Annual Prevalence of Norovirus Strains

To characterize the annual variations in predominant strains, the norovirus season was defined as the period beginning on July 1st and ending on June 30th of the following year. This places the peak of the norovirus season which occurs between November and March in the Northern hemisphere, at the center of the period [28].

### 2.8. Statistical Analysis

Statistical analyses were performed using GraphPad (Prism version 5.04) (IBM, Armonk, NY, USA). Data were summarized with frequencies and percentages for categorical variables and medians and interquartile ranges (IQR) for continuous variables as appropriate. Between-group (e.g., norovirus positive vs. negative) comparisons were performed using Chi-square and Mann–Whitney U tests for categorical and continuous data, respectively. We calculated two-tailed *p* values and set the significance level α at 0.05.

## 3. Results

### 3.1. Demographics and Clinical Characteristics

A total of 3,347 AGE cases were enrolled, of whom 26.9% (900/3347) tested positive for norovirus. Out of the 900 children who tested positive for norovirus, 41.0% of children had isolated vomiting (i.e., no diarrhea) prior to their ED visit. The median (IQR) age of the norovirus positive cases was 17.5 (10.8, 37.5) months and of the controls was 15.5 (6.7, 43.7) months; *p* = 0.72. The norovirus positivity rate was highest in children aged 1.0 to 3.0 years of age (393/1347, 29.2%; 95% CI: 18.7%, 34.3%) compared with younger (272/1028, 26.5%; 95% CI: 15.2%, 31.4%) and older (235/972, 24.2%; 95% CI 14.7%, 29.3%) children (Table 1).

### 3.2. Norovirus Genogroup and Genotype Distribution

Norovirus genogroup GI and GII accounted for 3.3% (31/900; 95% CI: 22–46) and 96.7% (872/900; 95% CI; 863–881) of all norovirus infections, respectively: two participants had G1/GII codetections. Although relatively similar in terms of vomiting and diarrhea frequency, children infected by norovirus GII were more likely to be febrile and to have a more prolonged illness at the time of ED presentation when compared to GI; Table 2. Only 17.1% (140/870) of norovirus GII cases presented with moderate to severe dehydration.

In the capsid region a total of 96.4% (869/900) samples were sequenced, of which 3.2% (28/869) belong to norovirus GI and 96.7% (841/869) belong to norovirus GII (Figure 1). Seventeen capsid based genotypes were identified, out of which seven were norovirus GI: GI.1 (25%; 7/28), GI.2 (10.7%; 3/28), GI.3 (28.6%; 8/28), GI.5 (3.6%; 1/28), GI.6 (17.9%; 5/28), and GI.7 (14.3%; 4/28). Eleven GII genotypes were identified, the most common of which were GII.4 (49.5%; 416/841), followed by GII.3 (28.3%; 238/841), GII.2 (11.2%; 94/841), GII.6 (6.8%; 57/841), and GII.7 (2.1%; 18/841) with the remaining genotypes contributed to 2.1% of the total genotypes characterized (Table 3). All 872 GII cases were sequenced in the RdRp region, of which 97.5% (851/872) could be typed into 11 different ORF-1 based genotypes. GII.P2 (N = 22; 2.6%), GII.P4 (N = 23; 2.7%), GII.P6 (N = 1; 0.1%), GII.P7 (N = 96; 11.3%), GII.P8 (N = 1; 0.1%), GII.P12 (N = 217; 25.5%), GII.P16 (N = 380; 38.8%), GII.P17 (N = 7; 0.8%), GII.P21 (N = 12; 1.4%), GII P3I (N = 141; 16.6%) and GIIP31/GII.P4 (N = 1; 0.1%).

Forty-one of the 47 (87.2%) norovirus positive control participants had their samples sequenced for RdRp region and 82.9% (39/47) for the capsid region. Seven RdRp based genotypes and 6 ORF-2 based genotypes were identified. The prevalent ORF-1 and ORF-2 strains identified were GII.P16 (46.3%; 19/41) and GII.2 (25.6%; 10/39).

#### Predominant Norovirus GII RdRp/Capsid Combination and Seasonal Distribution among AGE

For the annual seasonal periods of December 2014 to June 2015, July 2015–June 2016, July 2016–June 2017, and July 2017–June 2018, a total of 861 samples were sequenced of which 43 different combinations of RdRp and capsid genotypes were identified. The most prevalent strains were GII.4 Sydney[P16] (23.9%; 206/841), GII3[P12] (13.5%; 159/841), and GII.4 Sydney[P31] (13.8% 116/841). The remaining genotypes each contributed to < 10% of the infections. Predominant genotypes varied among seasonal years (Table 3).

The recombinant strain GII.4 Sydney [P31] was the predominant strain representing 52.3% (44/85) of all circulating GII strains from December 2014 to June 2015 and decreased subsequently (Table 3).

For the annual periods July 2015–June 2016 and July 2017–June 2018, recombinant strain GII.4 Sydney[P16] was the most common strain but it was only 26.7% (65/250) of the total norovirus cases during 2015/2016 period and 32.6% (105/268) during 2017/2018. During the intervening annual period of 2016–2017, the recombinant GII.4 disappeared and a non GII.4 strain recombinant strain GII.3[P12] was predominant and represented 38.8% (97/258) of all the total cases.

### 3.3. Difference in Clinical Features between GI and GII, Predominant Genotypes, and Lower Infective Genotypes

Of the 861 participants 841 (97.6%) participants were successfully contacted for follow-up and the 20 participants who were not able to be contacted for a follow up were excluded from further analysis. No significant difference was found in terms of the median (IQR) MVS total illness scores between GI 11.0 (9.0, 14.0) and GII 12.0 (10.0,14.0) genotypes, respectively; *p* = 0.51 (Table 2). Compared with all other GII cases, AGE cases with GII.4 had a longer duration of diarrhea (*p* = 0.002) and a greater maximum number of diarrheal stools per day (*p* < 0.001): Table 4. The duration of vomiting among GII.4 AGE was longer than those with other non-GII.4 genotypes (*p* < 0.001). The total illness median MVS score was higher among GII.4 infected children (12.0; IQR 10.0, 14.0) compared with other non-GII4 positive cases (11.0; IQR: 9.0, 13.0); *p* < 0.001.

## 4. Discussion

Our study has provided a comprehensive view of norovirus genogroups causing AGE in children in the province of Alberta, Canada. We demonstrated year-to-year variability in the predominant norovirus genotype; however, there was remarkable stability in terms of genogroup prevalence with GII accounting for 97% of all cases. Within the genogroup, GII.4 genotype was predominant in three of the four study years with GII.3 predominating in the one outlier year. While we identified some differences in the clinical characteristics of children infected by norovirus GI and GII genogroups, their overall illness severity scores were similar. However, within the GII genogroup, those with GII.4 infections had more severe overall illness as quantified by the MVS than those with other members of the GII genogroup.

GI accounted for 3.3% of all norovirus infections in our study, GII viruses in particular. GII.4 viruses are responsible for the majority of the norovirus outbreaks in people of all ages globally, whereas GI strains are more often noticed in foodborne and waterborne outbreaks. For example, the GI.6 virus that emerged in 2012 was more often associated with foodborne disease outbreaks than the GII.4 viruses, which are strongly associated with person-to-person transmission and outbreaks in health care settings, resulting in an increased risk of more-severe disease outcomes such as hospitalization and death compared to other GI and GII viruses [10].

Most prior evaluations of norovirus genotype prevalence have employed outbreak data or sporadic gastroenteritis samples captured at single hospital sites [10,29,30]. Our study enrolled a large number of children over four years at the province’s two tertiary care pediatric EDs as well as through a province-wide telephone triage help advice line. All children with AGE, regardless of disease severity, who consented to participate submitted specimens for testing. Notably we included children with isolated vomiting and were able to routinely detect pathogens, through the use of rectal swabs [31]. Including children with isolated vomiting and those not seeking care enabled our cohort to provide a unique and comprehensive picture of pediatric norovirus AGE.

Unlike most studies where partial capsid sequences were used for genotyping of norovirus, we used dual genotyping targeting the RdRp and capsid genes to genotype norovirus strains. Our data showed that strains GII.4 (49.5%), GII.3 (28.3%), and GII.2 (11.1%) were the three most common norovirus genotypes in our study. Our results are similar from those reported in pediatric patients from Hong Kong and Australia where strains belonging to the GII.4 cluster were most commonly found in sporadic gastroenteritis followed by strains belonging to the GII.3 genotype and other GII genotypes [32,33]. However, although predominant in our study, GII.4 prevalence was 10% lower than previous reports describing outbreaks in long-term care facilities and hospitals [34]. There may be several potential explanations for this conflicting finding. First, virus strains circulating and predominating may differ over time. While we report data from 2014 to 2018, the earlier reports were primarily from 2011 to 2014 [35,36,37,38]. Second, it is possible that host factors may affect population susceptibility to specific norovirus strains. Third, elderly populations and nursing home settings are prone to high GII.4 circulation and transmission [39]. Another possible explanation may be the recruitment of eligible AGE participants from a provincial telephone nursing advice line who were deemed not to require emergency treatment.

In the annual period of 2016–2017, GII.3 was the predominant genotype. Although the majority of AGE outbreaks due to norovirus infection are caused by GII.4 norovirus [40], GII.3 norovirus are one of the most common genotypes associated with sporadic norovirus infection, particularly in children, where they often are identified as the dominant genotype [41]. Interestingly, studies on historical stool specimens from the 1970 s and 1980 identified GII.3, rather than GII.4, as the main circulating norovirus genotype in hospitalized pediatric diarrhea [42]. Our results are in agreement with the recent reports that also identified GII.3 as predominating in young children and infants [29,40,41,43]. By sequencing both RdRp and capsid genes, GII.4 Sydney[P16] and GII.4 Sydney[P31] were identified as the predominate recombinant strains. These two genotypes were first detected in 2012 in major outbreaks in Australia [44], and thereafter became the predominant strains. These strains have also been isolated in the United States, Denmark, Japan, Scotland, and Canada [17,18,35,45,46,47]. The GII.2 genotype was the third most predominant cause of norovirus infection, accounting for 16% of all norovirus infections. Interestingly, we observed a sudden increase in GII.2[P16] during 2016–2017. Similar increases during that time period have been described in Germany, France, and China [48,49,50]. In fact, in China, this strain was identified in 79% of the outbreaks that occurred in 2016 [51]. Full-length analysis of GII.2 VP1 and RdRp regions indicated that amino acid substitution in partial RdRp in the GII.P16 enhanced polymerase kinetics which may have led to its surge in prevalence during the 2016 and 2017 seasons [52].

We also noted that GII.4 infected children had more severe illness, as reflected by higher MVS scores, than children with non-GII.4 strains suggesting this genotype is more virulent [53]. Previous data on this topic is scant but does include a cohort of Finnish children < 2 years of age who were recruited between 2000 and 2002. The authors reported that children infected with norovirus GII.4 had longer durations of diarrhea and vomiting than other norovirus genotypes, suggesting greater virulence of norovirus GII.4 [36]. The other report, which concluded that hospitalizations and deaths were more likely in outbreaks associated with GII.4 viruses, was a review of published norovirus outbreaks [10] and is not reflective of pediatric endemic disease.

These findings nonetheless reflect a consistent pattern. The GII.4 strain is shed at higher levels [54], is more likely to induce vomiting, longer duration of diarrhea, and cause more severe disease in children [30,36,55,56]. In outbreaks, GII.4 norovirus has been associated with higher attack rate [57] and more severe clinical presentation compared to other norovirus genotypes [30]. The predominance of norovirus GII.4 may be due to a higher evolution rate which has enabled the emergence of strains with greater human histo-blood group antigens (HBGAs), binding affinity [58,59]. On the other hand, norovirus GII.2 and GII.3 may have an evolutionary hindrance due to a less progressive polymerase and lower rate of evolution [60].

A potential confounding factor in studies of virulence of norovirus genotypes is pre-existing immunity which might ameliorate the clinical course of AGE caused by older norovirus genotypes. While the GII.4 genotype has been in circulation for several decades, this has not diminished its pathogenicity. Although our study population included young children, it is possible that many of them have previously experienced a norovirus infection as the median age of the first symptomatic infection of norovirus GI and GII has been reported to be as young as five and eight months, respectively [61]. Our report supports a previous Polish study [62] which reported a peak of norovirus infection rate in children aged between 1 to 3 years [62]. This finding could be attributed to the fact that children in this age range begin to explore more actively their surroundings which may lead to a greater exposure to norovirus which leads to a greater frequency of disease [63]. Furthermore, this is a period when children frequently place items in their mouth and they have limited awareness of hygienic practices and therefore are more frequently exposed [64].

A limitation of this study includes norovirus genotypes that were characterized using short sequence data. Furthermore, investigations focusing on whole genome sequencing of the circulating variants, which would provide greater insight into their evolution in the community, were not performed. Thus, we are unable to shed light on the origins of the norovirus genotypes we have identified. A minor limitation of this study is to recall the participants 14 days after their enrollment instead of daily symptoms log.

Inclusion of children with isolated vomiting and those who did not seek care dramatically enhances the generalizability and novelty of our findings. These results have important implications for those working on candidate vaccines [65,66,67,68,69] where an accurate assessment of genotype specific disease burden over time is crucial.

In conclusion, we identified GII.P4 Sydney[P16] as the overall predominant strain; however, annual peaks by GII.3[P12] and GII.4 Sydney[P31] also occurred. The GII.4 genotype was associated with the most severe clinical presentation. Continuous surveillance for the circulating genotypes and their associated severity is important to inform prevention strategies in the development of a vaccine.

## Figures and Tables

**Figure 1 microorganisms-08-01873-f001:**
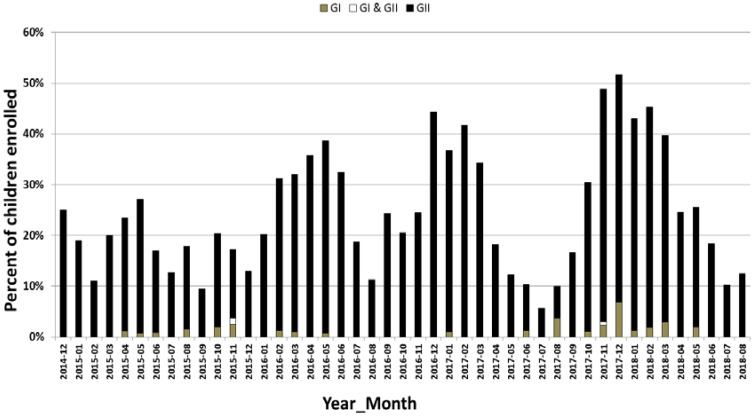
Monthly distribution of norovirus genogroup I and genogroup II in children with AGE from December 2014 to August 2018 in Alberta. The *y*-axis represents the percent of kids enrolled with AGE and the *x*-axis represents the month of the year. AGE: acute gastroenteritis.

**Table 1 microorganisms-08-01873-t001:** Demographic characteristics, enteropathogen codetection, and norovirus threshold cycle values of the study population.

	NoV-Positive Participants; N = 900 No. (%)	NoV-Negative Participants; N = 2447 No. (%)	*p*-Value
Sex			
Male; No. (%)	422 (46.9)	1328 (45.7)	0.56
Female; No. (%)	478 (53.1)	1119 (54.3)	
Age			
<1.0 year; No. (%)	272 (26.5)	756 (73.5)	0.03
1.0–3.0 years; No. (%)	393 (29.2)	954 (70.8)	
>3.0 years; No. (%)	235 (24.2)	737 (75.8)	
Year			
December 2014–December 2015	142 (15.8)	593 (24.2)	<0.001
January 2016–December 2016	318 (35.3)	775 (31.7)	
January 2017–December 2017	307 (34.1)	765 (31.3)	
January 2018–August 2018	133 (14.8)	314 (12.8)	
Number of Enteropathogens Detected			
0	661/899 (73.5)	989/2421 (40.9)	Not applicable
1	205/899 (22.8)	1233/2421 (50.9)	
2	31/899 (3.4)	187/2421 (7.7)	
3	1/899 (0.1)	12/2421 (0.5)	
4	1/899 (0.1)	0/2421 (0)	
Enrollment Location			
ED	668 (74.2)	2,028 (82.9)	<0.001
Health-Link Alberta	232 (25.8)	419 (17.1)	
Norovirus Ct Value			
GI (N = 31) Median (IQR)	22.8 (16.75, 31.4)	Not applicable	Not applicable
GII (N = 872) Median (IQR)	18.3 (15.7, 23.6)	Not applicable	Not applicable

Abbreviations: Nov, Norovirus GI, genogroup I; GII, genogroup II; ED, Emergency Department.

**Table 2 microorganisms-08-01873-t002:** Clinical characteristics of Norovirus GI and GII Genogroups.

	GI Positive N = 31	GII Positive N = 872	*p* Value
Age, Months, Median (IQR)	33.5 (17.3, 69.2)	17.3 (10.6, 36.1)	0.002
Fever, Yes, N (%)	11/30 (36.7)	348/823 (42.3)	0.58
Max temperature (°C), Median (IQR)	36.5 (36.5, 38.5)	36.5 (36.5, 38.6)	0.93
Fever, No, N (%)	19/30 (63.3)	475/823 (57.7)	0.58
Vomiting, Yes, N (%)	30/31 (96.8)	846/868 (97.5)	>0.99
Maximal Frequency (per 24 h), Median (IQR)	7.0 (4.0, 12.0)	7.0 (5.0, 11.0)	0.75
Frequency Past 24 h, Median (IQR)	7.0 (4.0, 10.0)	5.0 (3.0, 10.0)	0.25
Duration, hours, Median (IQR)	53.0 (15.2, 77.8)	64.3 (30.7, 114.4)	0.20
Diarrhea, Yes, N (%)	27/30 (90.0)	693/843 (82.2)	0.33
Maximal Frequency (per 24 h), Median (IQR)	3.0 (2.0, 5.0)	4.0 (2.0, 6.0)	0.60
Frequency Past 24 h, Median (IQR)	0 (0, 3.0)	1.0 (0, 4.0)	0.35
Duration, hours, Median (IQR)	48.0 (24.0, 93.0)	88.7 (24.0, 157.2)	0.09
Clinical Dehydration Scale score [27], Median (IQR)	2.0 (0, 3.0)	2.0 (1.0, 4.0)	0.17
No dehydration (score 0)	9 (29.0)	190 (21.8)	0.62
Mild (score 1–4)	18 (58.1)	531 (61.0)	
Moderate to Severe (score 5–8)	4 (12.9)	149 (17.1)	
MVS score at baseline	8.0 (7.0, 12.0)	9.0 (7.0, 11.0)	0.36
MVS score at follow-up phase	6.0 (3.0, 8.5)	6.0 (4.0, 9.0)	0.50
MVS score for total illness course	11.0 (9.0, 14.0)	12.0 (10.0, 14.0)	0.51

Abbreviations: MVS, Modified Vesikari Scale; GI, genogroup I; GII, genogroup II.

**Table 3 microorganisms-08-01873-t003:** Seasonal distribution of Norovirus Genogroups and Genotypes.

	December 2014– June 2015 No. (%)	July 2015– June 2016 No. (%)	July 2016– June 2017 No. (%)	July 2017– June 2018 No. (%)	Total No. (%)
TOTAL (N = 892)					
GI (N = 31; 3.5%)	4 (12.9)	8 (25.8)	3 (9.6)	16 (51.7)	
GI.1	0 (0.0)	0 (0)	2 (66.7)	5 (31.3)	7 (22.6)
G1.2	0(0.0)	2 (25.0)	0 (0)	1 (6.3)	3 (9.7)
GI.3	2 (50.0)	3 (37.5)	0 (0)	3 (18.6)	8 (25.8)
GI.5	0 (0)	1 (12.5)	0 (0)	0 (0)	1 (3.2)
GI.6	0 (0)	1 (12.5)	0 (0)	4 (25.0)	5 (16.1)
GI.7	1(25.0)	0 (0)	0 (0)	3	4 (12.9)
Genotype not determined	1(25.0)	1 (12.5)	1(33.3)	0	3 (9.7)
GII (N = 861; 96.5%)	85 (9.9)	250 (29.0)	258 (30.0)	268 (31.1)	
GII.2[P16]	0 (0)	0 (0)	60 (23.3)	2 (0.7)	62 (7.2)
GII 3					
GII.3[P12]	0(0)	32 (12.8)	97 * (37.6)	30 (11.2)	159 (18.5)
GII.3[P16]	0(0)	8 (3.2)	8 (3.1)	27 (10.1)	43 (5.0)
GII.4					
GII.4 Sydney[P12]	2 (2.4)	4 (1.6)	4 (1.6)	21 (7.8)	31(3.6)
GII.4 Sydney[P16]	0 (0)	65 * (26)	36 (14.0)	105 * (39.2)	206 (23.9)
GII.4 Sydney[P31]	44 * (51.7)	48 (19.2)	9 (3.5)	15 (5.6)	116 (13.5)
GII.4 Sydney[P4New Orleans]	1 (1.2)	8 (3.2)	12 (4.7)	1 (0.4)	22 (2.6)
GII.6[P7]	10 (11.8)	25 (10)	1 (0.4)	12 (4.5)	48 (5.6)
Other GII Genotypes	27 (31.7)	53 (21.2)	22 (8.5)	43 (16)	145 (16.80)
Genotype not determined	1 (12)	7 (2.8)	9 (3.5)	12 (4.5)	29 (3.4)

* Predominant strain; Abbreviations: GI, genogroup I; GII, genogroup II.

**Table 4 microorganisms-08-01873-t004:** Clinical characteristics of Norovirus GII.4 vs. non-GII.4 Positive Cases.

Variable	GII.4 N = 416	Non- GII.4 N = 425	*p*-value
Age, months, Median (IQR)	16.9 (11.4, 28.9)	17.7 (10.2, 46.8)	0.28
Diarrhea, yes, No. (%)	353/406 (86.9)	318 (77.8)	0.001
Diarrhea duration hours, Median (IQR)	110.8 (48.0, 175.1)	72.0 (9.3, 139.4)	<0.001
Maximum number of times/24-h period, Median (IQR)	4.0 (2.0, 7.0)	3.0 (1.0, 5.0)	<0.001
Vomiting, No. (%)	407/415 (98.1)	410/423 (96.9)	0.38
Vomiting duration hours, Median (IQR)	74.7 (38.0, 125.4)	53.8 (23.6, 103.0)	<0.001
Maximum number of times/24-h period, Median (IQR)	7 (4.0, 11.0)	7 (5.0, 10.0)	0.99
MVS score at baseline	9.0 (7.0, 11.0)	9.0 (7.0, 10.0)	0.005
MVS score at follow-up phase	7.0 (4.0, 10.0)	6.0 (4.0, 9.0)	0.002
MVS score for total illness course	12.0 (10.0, 14.0)	11.0 (9.0, 13.0)	<0.001

Abbreviations: GII, genogroup II.

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
