# Peer review of "Differences in Illness Severity among Circulating Norovirus Genotypes in a Large Pediatric Cohort with Acute Gastroenteritis"

_microorganisms, 2020, doi:10.3390/microorganisms8121873_

Round 1

Reviewer 1 Report

This is a large Canadian study that examined patterns of disease severity by norovirus genotype. The results have implications for which genotypes to include in future vaccines. Strengths include the large sample size of 900 norovirus positive cases. The findings support limited previous studies that suggest that GII.4 is associated with more severe disease.

ABSTRACT-- add the location where the children were captured (ED and nursing advice line)

METHODS

-p.3 line 95, section 2.3: children over 12 may provide more accurate information if they provide it themselves as compared to obtaining from caregiver

-Methods do not describe how lab testing was done for other enteropathogens and which other enteropathogens were tested

-p.3 line 131, section 2.6: in the case of cases captured through the nursing advise line, how was temperature obtained? Also, the description of the modified vesikari score does not mention dehydration, but this is included in the results. How was dehydration assessed in cases captured through the nursing advise line?

RESULTS

-p.4 line 157-158- wording of this sentence is confusing

-Table 1 should include the proportions recruited in the ED vs. from the nurse line

-Figure 1" Replace "kids" with "children"

-p.7 line 214: "and a GREATER maximum number"

DISCUSSION

-How do authors explain the low proportion of GI infections?

-Paragraph at line 269: consider including Bucardo, et al, (Infect Genet Evol 2017) who showed that over 16 years in Nicaragua, GII.4 was more likely to be detected in the hospital setting as compared to the outpatient or household, and was almost never detected in asymptomatic norovirus infections.

-A minor weakness that should be acknowledged in the reliance on recall of symptoms 2 weeks after enrollment vs. the use of daily symptom logs

-Paragraph at line 238: Another explanation for the lower GII.4 prevalence as compared to prior studies is the inclusion of less severe cases from the nurse advise line

Reviewer 2 Report

This is an interesting manuscript that reports differences in illness severity among circulating norovirus genotypes in a large pediatric cohort with acute gastroenteritis in Alberta, Canada.
The authors show that the predominant strain varied by year with GII.4 Sydney[P31]  predominant in 2014/15, GII.4 Sydney[P16] in 2015/16 and 2017/18 and GII.3[P12] in 2016/17. They also concluded that Genogroup II norovirus strains were predominant and clinical severity associated with the different strains varied with episodes being most severe among GII.4 genotype infected children.

The main strengths of the article are the large period 2014-2018 analyzed and the availability of a large number of samples, the clearness of the methods, results and discussion as well as the adequate number of tables. On the other hand, there are main aspects that should be improved like the mismatch between samples in the different tables:

In table 1 and table 2 there are 872 GII positive samples but in table 3 the GII samples decrease to 861. Finally the table 4 decrease to 841 positive samples for GII genogroup.

The authors should unify the data or give an explanation for this mismatches.

The paper also includes some little mistakes / misinformation:

  • There is a missing in the thousands separators (line 38, 106, 156, 161, table 1…)

  • The line 53 has words with various letter size.

  • In the Table 2, the fifth row for the GI genogroup is mistaken (19/30 (63.3)).

  • In the Table 3, the row nine for the 2017-2018 period has lose the percentage (18.6%).

  • There are some missed points in the genotype nomenclature (line 224, 225…).

  • In the line 306 the authors use GII4 genogroup instead GII genogroup.
